# Exploratory Behavioral Study of the Production and Processing of French Categorical Liaisons in Children with Expressive DLD

**DOI:** 10.3390/neurosci6040112

**Published:** 2025-11-06

**Authors:** Elisabeth Cesari, Bernard Laks, Frédéric Isel

**Affiliations:** 1UFR Littérature, Linguistique, Didactique (LLD), Institut de Linguistique et Phonétique Gé-nérales et Appliquées (ILPGA), Université Paris Nanterre, Modyco, 92000 Nanterre, France; elisabeth.cesari.7@gmail.com (E.C.); laks.bernard@wanadoo.fr (B.L.); 2Laboratoire de Phonétique et Phonologie, UMR7018, CNRS, UFR Littérature, Linguistique, Didactique (LLD), Institut de Linguistique et Phonétique Générales et Appliquées (ILPGA), Université Sorbonne Nouvelle, 75005 Paris, France

**Keywords:** neurodevelopmental disorders, expressive Developmental Language Disorder (DLD), categorical liaison, production, perception, working memory, inhibition, clinical marker

## Abstract

Categorical liaison—defined as the obligatory pronunciation of a latent word in the form of a final consonant when followed by a vowel as the initial word or a word beginning with a silent “h” (e.g., des‿ours [dezuʁs])—is a robust phonological phenomenon in French and an informative window into morphophonological development. This exploratory behavioral study investigates the dissociation between perception and production of categorical liaisons among 24 French-speaking children aged 6–10 years diagnosed with expressive Developmental Language Disorder (DLD). A battery of nine ad hoc tasks assessed perception and production across words, pseudowords, noun phrases, and sentences. Results showed that children with DLD performed comparably to typically developing peers in perceiving unrealized categorical liaisons but exhibited significantly more omissions in production, regardless of context or age. Production deficits correlated with reduced working memory and inhibitory control. These preliminary findings provide descriptive data that can inform the development of standardized assessment tools and generate hypotheses about the cognitive mechanisms underlying categorical liaison difficulties in DLD.

## 1. Introduction

In France, the estimated prevalence of Developmental Language Disorder (DLD)—clinically referred to as dysphasie—ranges from 2% to 3% among school-aged children [1,2,3]. Within this population, expressive DLD, defined by marked impairments in phonological encoding and/or syntactic formulation, accounts for a substantial subset, with approximately 1% of children exhibiting a severe expressive profile [4]. These national estimates are broadly aligned with international prevalence data (e.g., [5,6]), although cross-country comparisons may be influenced by diagnostic heterogeneity, differences in diagnostic thresholds, and disparities in clinical resources and assessment protocols. As observed in other neurodevelopmental disorders, sex differences are notable, with male-to-female ratios ranging from 2:1 to 3:1 in favor of boys.

A core diagnostic feature of DLD, especially in its expressive form, is the absence of concurrent sensory, cognitive, or psychiatric impairments, which helps isolate the linguistic deficit as the primary domain of dysfunction. Gérard [7], as well as Rapin and Allen [8], proposed a tripartite classification of DLD subtypes: (1) expressive DLD; (2) receptive DLD, which involves impaired verbal comprehension with relatively fluent—though often semantically inappropriate—output; and (3) mixed DLD, which combines both expressive and receptive impairments.

Given that expressive DLD is the most prevalent and well-defined subtype, the present study focuses specifically on this profile. Moreover, children with expressive DLD typically exhibit preserved semantic and pragmatic competencies, allowing them to engage meaningfully in experimental paradigms targeting morphosyntactic and phonological processing. This preserved competence makes expressive DLD a particularly informative model for studying morphophonological processing in development, because it allows researchers to isolate production and processing deficits without major confounds from comprehension or pragmatic impairments.

### 1.1. Clinical Approach

From a clinical perspective, expressive DLD is defined as a disorder manifesting at four levels of linguistic analysis: phonological, lexical, morphological, and syntactic.

At the phonological level, difficulties in producing and processing sounds are frequently documented in psycholinguistic studies [9,10]. Léonard [11] hypothesized that these difficulties might be due to under-specified phonological representations, leading to reduced phonological distinctiveness and greater susceptibility to misperception and production errors.

At the lexical level, research converges to show that children with expressive DLD aged 3;9 to 5;8 present a restricted vocabulary in comparison to their typically developing peers [12,13]. Furthermore, studies have reported that individuals with expressive DLD frequently experience delays and inaccuracies in word recall, in contrast to their typically developing counterparts [14]. As the smaller lexical stock in people with expressive DLD has been widely observed in different languages, it is considered a reliable marker of this language disorder [7].

At the morphological level, individuals with expressive DLD encounter significant challenges in the formation of morphologically complex words [15]. Verbal morphology is frequently disrupted, particularly with respect to verb agreement [16].

At the syntactic level, children with expressive DLD produce few functional words, such as articles and prepositions, which lends a “telegraphic” style to their productions (e.g., “bird in tree”) and leads to dyssyntax (e.g., “On commence plus à avoir place” instead of “On commence à ne plus avoir de place”).

In contrast, semantics is typically preserved, as reported by Leonard [11]. This relative preservation of meaning makes expressive DLD particularly informative for studies of morphophonological processing, since deficits can be studied without major confounds from semantic impairment. It should also be noted that fully disentangling the respective contributions of phonological and syntactic impairments may require neurophysiological approaches such as electroencephalography (EEG), which offers fine-grained temporal resolution of language processing (see [17] for an overview).

### 1.2. Linguistic Perspective

Categorical liaison is defined as the phonological phenomenon whereby a normally silent final consonant is pronounced when the following word begins with a vowel or a silent “h” (e.g., un‿arbre [$\tilde{œn}$aʁbʁ]).

The liaison is categorically obligatory in specific cases where word 1 is syntactically dependent on word 2, i.e., it would not be correct to use the second term without the first:Between a determiner and its initial vowel noun (un, les, des, mon, son, ton, mes, tes, ses → un‿ours “a bear”);Between a determiner and an adjective (les‿autres enfants “the other children”);Between a clitic pronoun and a verb, or between two clitics (ils‿ont, je les‿aime, elles‿en ont);In the case of postposed personal pronouns (vient‿il?);In the case of enclitics, forming a single prosodic unit with the preceding term (de quoi parle-t-il ?);In specific fixed expressions and compound words (de temps‿en temps, porc‿épic, tout‿à fait).

In this study, we focused primarily on determiner–noun contexts, which represent the prototypical and most frequent cases of obligatory liaison, while also including more complex frames such as subject–verb–determiner–noun or pseudoword sentences in the perception tasks. This choice was motivated by their early acquisition in neurotypical children, their high frequency in spontaneous speech, and their suitability for controlled experimental design. Other obligatory contexts (e.g., enclitics, fixed expressions) were not included, and optional or illegal liaisons were deliberately excluded to avoid ambiguity.

As a mandatory property, categorical liaison precludes avoidance strategies: in the relevant contexts, speakers cannot simply “opt out” of producing the liaison without violating a grammatical norm. By contrast with optional liaison (e.g., mais (‿) ils), obligatory liaison minimizes diachronic, regional, stylistic, and socio-cultural variability [18,19,20]. For this reason, our design focused exclusively on obligatory cases.

In turn, “impossible” (illegal) liaison—such as before words with h aspiré (un # hangar, les # héros)—was excluded because it introduces orthography–phonology conflicts and may alter morphosyntactic parsing and thus meaning. For example, strings like un savant aveugle … can be parsed either as [NP + adjective] (“a blind scientist”) or, in sentential contexts, as [NP + verb] (“a scientist blinds …”), an ambiguity that is orthogonal to our research question.

Categorical liaison is of particular interest to psycholinguistics because it is a multifaceted phenomenon: it involves phonological activation (pronunciation of the latent consonant), morphosyntactic integration (recognition of dependency between two words), and prosodic planning (resyllabification across word boundaries). Studying its acquisition thus offers insight into how children coordinate multiple levels of linguistic representation.

The challenge in producing categorical liaisons, as evidenced in both children and adults with expressive DLD, is striking, given that such liaisons are generally mastered without difficulty by neurotypical children from the age of 4 [18,19,20,21]. Another characteristic of expressive DLD is the imbalance between production and perception capacities: while production of categorical liaisons is severely impaired, perception is relatively preserved, though not identical to neurotypical performance [22,23,24].

### 1.3. Psycholinguistic Models of Categorical Liaison Acquisition

In the psycholinguistic literature, two benchmark models have been proposed to account for the acquisition of categorical liaison in French.

Firstly, the phonological model of Wauquier-Gravelines and Braud [25] which relies on the speaker’s “innate” knowledge of the phonological structure of their language. This model hypothesizes that the language learner stores all possible word variations with or without a liaison, e.g., des [z]amis, un [n]ami, petit [p]etit ami.

In contrast, the constructionist model proposed by Chevrot, Dugua and Fayol [26] assumes that the child gradually extracts patterns from frequently heard categorical liaisons, generalizing them to new items. Individuals with expressive DLD do not systematically link the two words involved in this juncture phenomenon to form a single phonologically perceptible word, even in categorical contexts [22,23,24]. Initially, the child gradually extracts patterns from frequently heard categorical liaisons. Taking the example of the word/ami/in different contexts, the child associates the plural with the pronunciation of the liaison consonant [z] ([z]amis) and the singular with the liaison consonant [n] (un [n]ami). Secondly, the extraction of regularities will lead to the implementation of a generalization procedure which will enable the child to apply these rules (e.g., [DETERMINANT—z—[vowel]—NAME] in the plural or DETERMINANT—n—[vowel]—NAME] in the singular) to any liaison without necessarily having heard it beforehand.

### 1.4. Cognitive Theories of DLD

Several cognitive theories have been proposed to explain DLD, particularly its expressive forms. One influential hypothesis attributes the disorder to deficits in working memory, specifically the phonological loop [25,26,27,28].

In addition, Ullman and colleagues [29,30,31,32] have proposed that procedural memory—responsible for implicit learning of rule-based sequences—is impaired in DLD, while declarative memory remains relatively intact and compensatory [33,34,35].

Because categorical liaison requires rapid integration of phonological and syntactic information, it provides a sensitive test case for evaluating whether procedural memory deficits contribute to morphophonological difficulties in DLD.

### 1.5. Rationale and Aim of This Research

The objective of this research endeavor was to enhance the comprehension of language impairment in children diagnosed with phonological–syntactic expressive DLD who encounter the categorical liaison in French. The present study focuses on the frequency and persistence of the omission error, defined as the failure to make the liaison despite its obligatory nature, contingent on the linguistic context in which the liaison is anticipated and the lexical frequency of the words to which the liaison is applied.

To our knowledge, no fully standardized or validated battery currently exists for assessing categorical liaison production and perception in children with DLD. In designing our protocol, we drew inspiration from established tasks in the literature (e.g., those of Chevrot and colleagues and others in the field), adapting and extending them to suit our target populations and linguistic contexts. Consequently, we developed an ad hoc set of nine tasks designed to probe multiple linguistic contexts (words, pseudowords, noun phrases, and sentences) and to sample liaison production and perception under both structured and semi-spontaneous conditions.

The goal of employing such a battery was not to produce a diagnostic instrument or to provide confirmatory evidence for a particular theoretical model, but rather to generate descriptive data that may inform future tool development and hypothesis-driven research. By combining adapted tasks from prior work with novel stimuli, we aimed to maintain clinical feasibility, age appropriateness, and ecological validity.

Nevertheless, the absence of full validation, pilot reliability measures, and normative data remains a methodological limitation. The findings must therefore be interpreted cautiously as hypothesis-generating rather than definitive.

### 1.6. Predictions

In light of prior findings and given that our material primarily targeted determiner–noun contexts while also including more complex sentence frames (subject–verb–determiner–noun and pseudowords), we formulated the following exploratory expectations.

(1)Compared to typically developing peers, children with expressive DLD were expected to produce the canonical (unlinked) form more frequently—i.e., omit categorical liaison–across all tested contexts. This tendency was anticipated to be relatively stable across the 6–10 age range (pre-readers vs. readers), rather than showing a strong age-related improvement.(2)The error profile was expected to be dominated by omissions, with comparatively fewer substitutions/additions/regularizations. We explored whether word length and lexical frequency would modulate omission rates but did not expect large effects under our task constraints.(3)Individual differences in executive resources were expected to modulate production, such that higher working-memory span and better inhibitory control (indexed by digit span and a flanker-type measure) would be associated with fewer omissions.(4)For receptive skills, we anticipated relatively preserved detection of unrealized categorical liaisons in children with DLD, albeit at a slightly lower level than in typically developing children.(5)Finally, we explored a potential influence of literacy (readers vs. pre-readers), expecting at most modest benefits (e.g., in perception or in high-frequency contexts) without eliminating group differences. These predictions are descriptive and hypothesis-generating; they are intended to guide exploratory analyses rather than to test a fully specified confirmatory model.

## 2. Materials and Methods

### 2.1. Participants

A total of 24 participants diagnosed with expressive DLD (17 males and 7 females, confirming the prevalence of this disorder in the male population) and 22 neurotypical children (14 males and 8 females), all native speakers of French, aged between 6 and 10 years, were included in the study. The distribution of participants at each age is displayed in Table 1 for both groups of children.

#### 2.1.1. Children with Expressive DLD

A total of twenty-four children, ranging in age from 6 to 10 years, have been diagnosed with expressive DLD. The children are registered with the Maison Départementale des Personnes Handicapées (MDPH—Departmental House for the Disabled) of the Bas Rhin (69) and Val d’Oise (95) departments in France. The 24 children who were recruited to the study had no other declared pathology. All of them were enrolled in two ULIS (Unités Localisées pour l’Inclusion Scolaire, Local Units for School Inclusion) schemes, which enable pupils with disabilities to attend mainstream schools. The level of education of the children with expressive DLD was difficult to assess accurately due to the inclusive schooling and the special needs of the children. However, on average, they are generally two to three years behind the expected school level for their age. As previously stated, the linguistic impairments experienced by children with expressive DLD encompass difficulties with sounds (e.g., discrimination, manipulation, syllable alteration), reduced lexicon, agrammatism and dyssyntaxia, which collectively impair oral expression, reading and written production. It is noteworthy that the skills of children with expressive DLD exhibit significant inter-individual variability due to the heterogeneity of their abilities.

#### 2.1.2. Neurotypical Children

The twenty-two neurotypical children who made up the control group were also selected from state schools in the Hauts de Seine (92) and Val d’Oise (95) departments in France, with the aim of achieving a similar distribution of participants in the control group as in the dysphasic group. The selection was made in agreement with the teachers, and the children had the expected school level, with no difficulties in either French or mathematics. None of them had repeated or skipped a grade. Prior to participation in the study, the parents were informed individually of the objectives of the study, as well as of the experimental protocol and the procedure for storing and anonymizing the data. They then gave their written consent. The data collected was anonymized by applying the European Data FAIR principle [36] in collaboration with the HumaNum TGIR (https://www.huma-num.fr, accessed on 30 April 2022) for experimental data management. Data processing was carried out in accordance with the General Data Protection Regulation (Règlement général de protection des données (RGPD) in French) Centre National de la Recherche Scientifique (CNRS). The research protocol was reviewed and approved by the Scientific Committee of the MoDyCo Laboratory (CNRS, Université Paris Nanterre), in accordance with the ethical standards and regulations in place at that time, and in full compliance with the Declaration of Helsinki. In addition, the study received authorization from the school administration of the ULIS program (a specialized inclusion unit under the supervision of the French Ministry of National Education), where the data collection took place.

### 2.2. Experimental Materials: Verbal and Non-Verbal Tests

This study was designed as an exploratory cross-sectional investigation, aimed at generating descriptive data rather than testing a fully confirmatory model. Because no standardized or validated battery currently exists for assessing categorical liaison in children with Developmental Language Disorder (DLD), we implemented an ad hoc protocol combining production and perception tasks. This approach allowed us to sample categorical liaison performance across multiple contexts (words, pseudowords, noun phrases, and short sentences) and in a range of elicitation conditions (structured, semi-directed, and spontaneous).

A series of nine verbal and one non-verbal tests were administered, encompassing tasks ranging from word reading to text reading, as well as guided conversation and free interview. Table 2 provides a summary of these tests.

The tasks were partly adapted from previous studies [22,26,28] and partly designed specifically for this project in order to make them age-appropriate and accessible for children with expressive DLD. This diversity of tasks was a deliberate choice to maximize ecological validity and maintain children’s engagement. However, it also introduces some uncontrolled variability, and no formal reliability coefficients could be computed. For this reason, the findings should be interpreted as preliminary and hypothesis-generating, serving as groundwork for the future development and validation of standardized tools.

The verbal tests of perception consisted of two tasks. In the first task, the child was asked to repeat only one of three spoken variations in a nominal phrase composed of a determiner and a noun, with a change in the liaison (Test 1). Six items were presented. The same modality was used for a 12-item exercise constructed with subject–verb–determiner–noun/logatom (pseudoword) sentences (Test 2). In the perception tasks, omission errors are deliberately inserted to assess the child’s ability to recognize the omission as an error or a correct formulation.The verbal tests of production comprised two denomination tests that were adapted from previous studies [22,28]. The child was presented with an image and was required to state the element and the number of representatives of that element they could see. In order to vary the measurement methods, the following modifications were made: (1) a counting exercise was supported by an illustration with the aim of attenuating the impact of anomia and word confusion, and minimizing the task in working memory; (2) a series of five riddles was used to stimulate spontaneity and to reinforce the determiner/name liaison; (3) a story to be told based on six drawings required the use of nouns beginning with a vowel to be repeated several times, playing on the number. Figure 1 presents a series of six drawings used to encourage children to produce categorical liaisons.

The utilization of concise sentences, monosyllabic and plurisyllabic words that were straightforward to pronounce and semantically uncomplicated ensured the comprehensibility of the material. The liaison was presented in a variety of ways, ranging from naming to counting to a guessing game, in order to preserve the spontaneity of the participants and avoid any awareness of the research purpose. The context of production was semi-directed or spontaneous. The words used in the verbal tests were selected in Open Lexicon [37] and all had a high frequency of occurrence. The participants performed a memory span exercise to test immediate memory and a reverse span exercise to test working memory.

### 2.3. Procedure

The administration of the tests was conducted by teachers and speech therapists who were regularly engaged with the children, thereby ensuring that the method was neither intrusive nor disturbed by an assessment situation that could alter the spontaneous nature of the responses. Indeed, the speech and language therapy report of two of the children with expressive DLD expresses a very marked fear of assessment situations and a refusal of any assessment situation. It was determined that a playful approach alone was insufficient to instill confidence in a child with expressive DLD, who often finds themself in evaluation situations. The test was administered during a session or after class and lasted between 20 and 30 min. The children were not informed of the liaison, yet they were aware that their responses were being recorded. The transcribed recordings were then analyzed for the purpose of data collection and analysis.

### 2.4. Data Analysis

Due to the heterogeneity of the age distribution, the level of impairment of the participants with expressive DLD and the size of the sample (less than 30 participants), we chose to use the non-parametric inferential Mann–Whitney test (MW; [38]) using JASP 0.16.1 software (http://www.jasp-stats.org, accessed on 17 February 2022; [39]). A total of 78 categorical liaisons were expected, including 18 for the perceptual tests. Responses were categorized as correct or incorrect. In perception, an error of omission occurs when the child accepts the form without a liaison as correct when the liaison is required. In production, an error of omission is characterized by the absence of a categorical liaison when the liaison is required. The dependent variables were the different types of scores expressed as percentages. In order to validate the score for the memory span exercises, it is necessary to complete two exercises with the same number of items. The score used is the last number of digits returned on the last correct trial.

Selective attention, as measured by the Flanker Test, was calculated as the percentage of correct arrow direction choices for twenty screen presentations. The reaction time to select the correct arrow on the keyboard after each new presentation was expressed in milliseconds.

## 3. Results

For statistical reporting, not significant (ns) = *p* > 0.10; marginal (mg) = 0.05 < *p* < 0.10; 0.01 < * *p* < 0.05; 0.001 < ** *p* < 0.01; *** *p* < 0.001.

### 3.1. Average Percentages of Correct Responses in Perception and Production

On average, the children with expressive DLD produced fewer categorical liaisons (M = 29.3%; SD = 26.4) than those of the control group (M = 85.9%; SD = 13). A Mann–Whitney U-Test showed that this difference was statistically significant (U = 504.00, *p* < 0.001, r = 0.9). The effect size (56.6%) is large (r = 0.9 > 0.5). The Mann–Whitney U-Test also indicated a significant difference in the percentage of correct perception of the categorical liaison between children with expressive DLD and neurotypical children (U = 446.00, *p* < 0.001; r = 0.7).

This effect indicates that the average perception rate was significantly smaller in the dysphasic group (M = 67.1%; SD = 20.7) than in the control group (M = 86.6%; SD = 8.5). The effect is large (19.5%; r = 0.7 > 0.5). Figure 2 displays the average percentages of correct responses in perception and production.

### 3.2. Comparison Between Children with Expressive DLD and Neurotypical Children’s Pre-Readers and Readers

The groups were divided according to age, which corresponds to the age at which children acquire reading skills. Table 3 presents the means and the standard deviations of correct perception and production of categorical liaisons for both pre-readers and readers with expressive DLD and neurotypical children.

Group of 6–7-year-olds (*N* =17 pre-readers)

The children with expressive DLD produced significantly fewer correct categorical liaisons (M = 17.6%, SD = 13.1%) than neurotypical children in the control group (M = 80.5%, SD = 14.0%). The size of the effect is 62.9% (MW = 72.000; *p* < 0.001). Contrastively, in perception, the main effect of Group was only marginally significant (MW = 54,000; *p* = 0.082), indicating that the children with expressive DLD performed as well as the neurotypical children.

Group of 8–10-year-olds (*N* = 29 readers)

While the perception performances did not significantly differ between the two groups (MW = 183.000; *p* > 0.10), the average percentage of correct production was significantly lower in the dysphasic group (M = 36.4%, SD = 30.0%) than in the control group (M = 89.0%, SD = 11.8%). The size effect is 52.6% (MW = 198.500; *p* < 0.001).

Finally, the readers with expressive DLD did not produce a significantly higher number of correct categorical liaisons (M = 36.4%; SD = 30.0%) than the pre-readers with expressive DLD (M = 17.6%; SD = 13.1%) (*p* > 0.10).

### 3.3. Correlation Between Perception and Production Tests

The correlation between the rate of categorical liaison’s perception and the rate of categorical liaison’s production was significant (r = 0.7; *p* < 0.001). This correlation was stronger in neurotypical children (r = 0.6; *p* = 0.008) than in children with expressive DLD (r = 0.5; *p* = 0.018).

### 3.4. Omission Error Rate Variations

As a reminder, in perception, an error of omission is the acceptance of the form as correct without a liaison when the liaison is required. In production, an error of omission is the absence of a categorical liaison when a liaison is required. Figure 3 displays the average errors of omission in the perception and production of the categorical liaison for both children with expressive DLD and neurotypical children. In perception, the Mann–Whitney U-test showed that the average percentage of forms judged correct when the mandatory liaison is omitted was significantly higher in children with expressive DLD (M = 23.51%; SD = 12%) than in neurotypical children (M = 7.57%; SD = 13%) (MW = 85.00, *p* < 0.001).

In production, the average error of omission percentage was significantly higher in children with expressive DLD (M = 63.11%; SD = 25.96%) than in children without expressive DLD (M = 9.75%; SD = 12.3%) (MW = 18.50, *p* < 0.001, r = 0.93 > 0.5).

### 3.5. Correlation Between Memory Span, Reverse Span and Production of the Categorical Liaison

The relationship between memory span and the rate of correct production of the categorical liaison calculated on all children with and without expressive DLD was significant (r = 0.6; *p* < 0.001). This correlation means that the higher the memory span score, the higher the rate of correct production of the categorical liaison. Similarly, the relationship between the reverse span and the rate of correct production of the categorical liaison was also significant (r = 0.4; *p* = 0.003). The more children were able to repeat long sequences of numbers backward, the better their production of categorical liaisons.

### 3.6. Correlations Between Flanker Task Scores and Perception and Production Scores

Table 4 summarizes the correlations between the Flanker task scores and perception and production scores for all children and for children with expressive DLD, respectively.

## 4. Discussion

### 4.1. Summary of Main Findings

This exploratory cross-sectional study provides preliminary evidence for a dissociation between perception and production of categorical liaison in children with expressive DLD. In line with previous findings [25,26,40], our results suggest that typically developing children have largely mastered the production and processing of categorical liaisons by age six, although production accuracy does not reach 100%. In contrast, children with expressive DLD displayed relatively good sensitivity to categorical liaison in perception (67.1%)–albeit significantly lower than their neurotypical peers (86.6%)–which suggests that phonological information about liaison is indeed represented in their lexicon rather than reflecting a primary perceptual deficit [25,26,27,41].

In production, however, children with DLD exhibited a markedly higher rate of omission errors across all contexts (words, pseudowords, noun phrases, and sentences). Performance did not improve significantly with age between 6 and 10 years, indicating a persistent production deficit that may not resolve spontaneously with increased linguistic exposure or reading acquisition. As observed in prior studies [25,26,27,42], omission was the predominant error type, raising the question of why children with DLD fail to produce categorical liaisons despite being able to perceive them.

### 4.2. Relation to Previous Literature and Cognitive Mechanisms

Several theoretical accounts have been proposed to explain the predominance of omissions. Wauquier [23] suggested a link between omission rate and working memory load, predicting that short and frequent words should yield fewer omissions than longer, less frequent words (e.g., les ours [lezuʁs] vs. les ingénieurs [lez$\tilde{ɛʒ}$enjœʁ]). Although we observed a significant correlation between working memory span and production performance, our data did not support a robust word-length effect: children with expressive DLD did not produce liaisons more accurately with short words than with long words, whether in noun phrases or sentences. This finding suggests that while working memory capacity is indeed involved, its influence is not limited to simple load effects such as word length but may relate to the quality of representational maintenance and selection during morphophonological planning.

Dugua and Chevrot [40] offered an alternative interpretation, suggesting that omission errors may reflect the absence of an available constructional scheme. In this view, the child fails to transition from memorizing lexical chunks to generating abstract productive schemas. Our findings are consistent with this interpretation: no shift was observed from omission errors toward substitution or addition errors that might indicate partial schema construction. This pattern is compatible with Ullman and Pierpont’s procedural deficit hypothesis [30], which attributes morphosyntactic difficulties in DLD to a dysfunction in procedural memory that limits the automatization of grammatical rules. Importantly, our results indicate that frequency of occurrence alone is insufficient to induce automatic, stable production of categorical liaison, as no frequency-driven improvement was observed across contexts.

The children’s reliance on the canonical form (liaison omission) is intriguing, given that the correct liaison form is typically acquired by neurotypical children around age 4 [26,42]. We hypothesize that children with DLD may fail to suppress or update an early-acquired canonical representation in their mental lexicon, leading to the persistent use of the unlinked form. This hypothesis is consistent with the random distribution and entrenchment of omission errors, pointing to difficulties with inhibitory control.

Our results on the Flanker task revealed slower reaction times and lower accuracy in children with DLD compared to neurotypical peers, a pattern consistent with findings reported by Larson et al. [43]. This pattern suggests a reduced efficiency in conflict monitoring and selective inhibition, rather than a generalized tendency toward impulsive responding. In other words, these children are not simply responding too quickly or acting impulsively, but seem to require more time and cognitive resources to resolve interference from distractors, resulting in both slower and less accurate responses.

This finding aligns with the view that inhibitory control difficulties may contribute to the failure to inhibit the canonical form and to update morphophonological representations. It also complements our working-memory results, as inefficient inhibition may interfere with the selection and maintenance of relevant representations in memory, further hindering schema formation. Taken together, these findings reinforce the idea that persistent categorical liaison omissions in expressive DLD reflect a broader interaction between procedural memory deficits, working memory limitations, and reduced inhibitory efficiency.

Importantly, performance in perception remained higher than production, which could reflect compensatory reliance on declarative memory during perception tasks and the attentional boost elicited by explicit questioning [44]. These findings collectively suggest that categorical liaison difficulties in DLD are multifactorial and cannot be explained solely by perceptual or frequency-based accounts. Instead, they support a model implicating procedural memory, working memory, and inhibitory control in the acquisition and stabilization of morphophonological schemas.

### 4.3. Theoretical and Clinical Implications

From a theoretical perspective, our results should be interpreted as hypothesis-generating rather than confirmatory. They raise new questions about how procedural and executive deficits interact to disrupt the transition from lexical chunk storage to productive schema formation. They also prompt investigation of whether declarative memory can compensate for procedural deficits in morphophonological learning and under which conditions such compensation might occur. These insights are relevant for refining usage-based and constructionist models of morphosyntactic development, which may need to integrate executive function constraints more explicitly.

Clinically, our findings support the idea that categorical liaison production could serve as a sensitive clinical indicator of expressive DLD beyond age five. Including categorical liaison tasks in language assessment batteries could help differentiate DLD from other profiles, such as late talkers or children with purely phonological delays. Moreover, the observed link between omission rates and executive function measures suggests that interventions targeting working memory and inhibitory control—combined with explicit morphophonological training—may be a promising direction for remediation.

### 4.4. Limitations and Future Directions

This study should be regarded as exploratory, and its findings interpreted as preliminary groundwork rather than conclusive evidence. Several limitations must be acknowledged:Small sample size (*N* = 24) restricted statistical power and the ability to detect subtle effects of age or linguistic context.The cross-sectional design prevented us from establishing developmental trajectories or causality.Absence of randomization or counterbalancing of task order may have introduced fatigue or order effects.The use of nine ad hoc tasks was necessary given the lack of standardized tools but introduced methodological heterogeneity.The Flanker task is not specifically normed for this population, which limits the interpretability of inhibitory control findings.Our reliance on nonparametric statistics (Mann–Whitney U tests, correlations) restricts the depth of inference regarding theoretical models.

Future research should replicate these findings with larger and more diverse cohorts, adopt longitudinal designs to track developmental change, and include additional tests targeting procedural memory and inhibitory control at both linguistic and motor levels. Experimental training studies should explore whether targeted interventions improve liaison production and whether such improvements generalize to spontaneous speech. Moreover, the development and validation of standardized assessment tools for categorical liaison will be essential to strengthen future research and clinical practice.

## 5. Conclusions

In summary, this exploratory study provides converging evidence that categorical liaison production remains persistently impaired in children with expressive DLD despite relatively preserved perception. The results highlight the complex interplay between procedural memory, working memory, and inhibitory control in morphophonological learning. Although preliminary, these findings contribute valuable descriptive data and lay the groundwork for future longitudinal and intervention studies that will ultimately determine the diagnostic and therapeutic utility of categorical liaison in expressive DLD.

In the long term, this line of research may contribute to earlier and more accurate identification of expressive DLD and to the development of targeted, evidence-based interventions that improve language outcomes.

## Figures and Tables

**Figure 1 neurosci-06-00112-f001:**
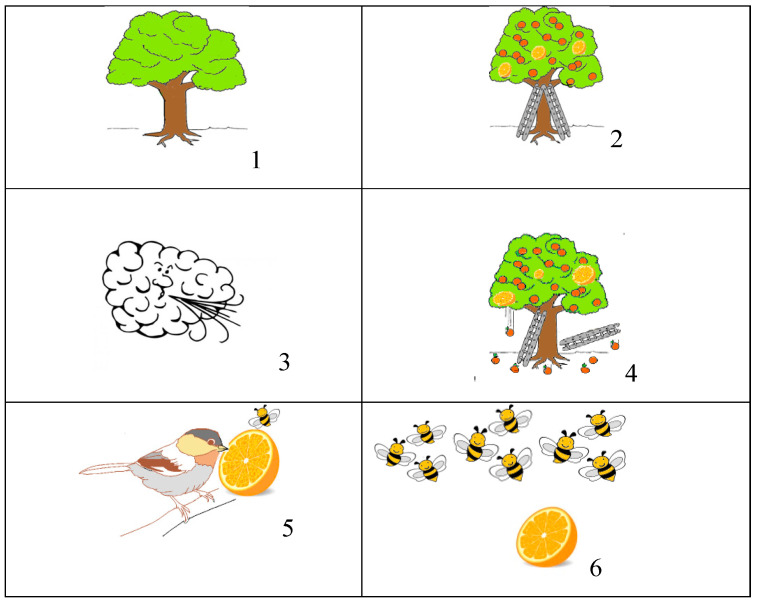
A series of six drawings used in one production task.

**Figure 2 neurosci-06-00112-f002:**
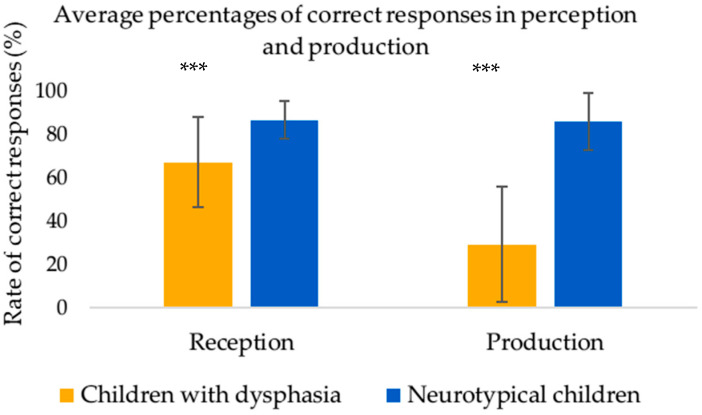
Average percentages of correct responses in perception and production. *** *p* < 0.001.

**Figure 3 neurosci-06-00112-f003:**
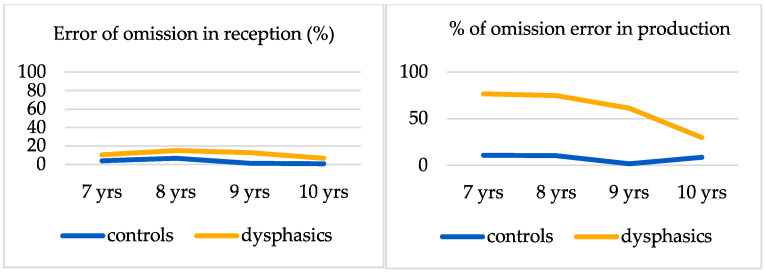
Errors of omission in the perception and production of the categorical liaison for children with expressive DLD and neurotypical children.

**Table 1 neurosci-06-00112-t001:** List of participants by age.

Age (In Yrs.)	Number of Neurotypical Children	Number of Children with Expressive DLD
6	2	2
7	6	7
8	5	6
9	3	4
10	6	5
Total	22	24

**Table 2 neurosci-06-00112-t002:** Summary of the various tests used.

Type of Test	Perception	Production
Verbal test	Detection of liaison	Denomination
	Memory span	Enumeration
		Counting
		Riddle
		Picture story to tell
		Guided conversation
		Reverse memory span
Nonverbal test	NA	Flanker test

**Table 3 neurosci-06-00112-t003:** Means and standard deviations of correct perception and production of categorical liaisons for both pre-readers and readers in children with expressive DLD and in neurotypical children.

	Perception (%)			Production (%)		
	Dys	Cont	Diff	Dys	Cont	Diff
Pre-reader	64.8 (26.6)	84.03 (4.6)	−19.2	17.6 (13.1)	80.5 (14.0)	−62.9
Reader	68.5 (17.0)	88.1 (10.0)	−19.6	36.4 (30.0)	89.0 (11.8)	−52.6
Difference	+3.7	+4.1		+18.8	+8.5	

Dys = dysphasic; Cont = control; Diff = difference.

**Table 4 neurosci-06-00112-t004:** The correlations between the Flanker task and selection attention scores and perception and production scores for both dysphasic and neurotypical children.

	Perception		Production	
	Dys	Cont	Dys	Cont
Selec. Attention	−0.05	+0.24	+0.56	+0.54 **
Flanker effect	−0.40	−0.30	−0.41 *	−0.28

Selec. attention = selective attention; * *p* < 0.05, ** *p* < 0.01.

## Data Availability

Data are unavailable due to privacy restrictions.

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
