# Peer review of "Exploratory Behavioral Study of the Production and Processing of French Categorical Liaisons in Children with Expressive DLD"

_neurosci, 2025, doi:10.3390/neurosci6040112_

Round 1

Reviewer 1 Report (Previous Reviewer 2)

Comments and Suggestions for Authors

I thank the authors for their thoughtful and detailed revisions. The manuscript is clearer, and the limitations of the design are now acknowledged more explicitly. I also appreciate the expanded introduction and methods section, and the effort made to respond constructively to the earlier critique.

However, several important methodological issues remain unresolved:

  1. Instrument validation – The tasks used remain unvalidated, with no pilot testing or reliability evidence provided.
  2. Task variability – The inclusion of multiple task formats (naming, enumeration, riddles, storytelling, guided conversation) introduces uncontrolled variability that is not sufficiently addressed.
  3. Cognitive measures – The Flanker task lacks calibration for this age group and clinical population. Although this is acknowledged, it still limits the strength of the findings.
  4. Design – There is still no randomization or counterbalancing of task administration, raising concerns about fatigue and order effects.
  5. Statistical analysis – The reliance on Mann–Whitney U tests and correlations remains a limitation, particularly given the ambitious theoretical claims concerning declarative/procedural memory and schema acquisition.

Because these issues are not resolved, the study cannot be considered definitive evidence for its theoretical claims. That said, the paper could have value if positioned more carefully as an exploratory contribution. I therefore recommend major revision, with the following conditions:

  • Reframe the manuscript explicitly as exploratory/pilot in the title, abstract, and discussion, ensuring that the contribution is presented as preliminary groundwork rather than conclusive evidence.
  • Emphasize methodological limitations early in the Introduction as well as in the Discussion. A short statement at the end of the Introduction should acknowledge the absence of validated tools, the lack of task standardization, and the exploratory nature of the design. This will help set the correct expectations for readers.
  • Moderate the theoretical claims. Findings should be presented as descriptive and hypothesis-generating rather than as confirmatory evidence for specific cognitive models.
  • Clarify the scope of the results: they provide useful descriptive data and point to areas for tool development, but they are not sufficient for strong causal or model-level conclusions.

If these changes are made, the manuscript will be more accurately positioned and may serve as a useful pilot contribution in this area of research.

Author Response

We sincerely thank Reviewer 1 for their constructive and insightful feedback. These comments were extremely helpful in clarifying the scope of our contribution and strengthening the framing of the study. The revisions made ensure that the manuscript is now more accurately positioned and may serve as a valuable pilot contribution to research on categorical liaison in children with DLD. Below, we provide a detailed, point-by-point response to each of the reviewer’s comments, highlighting the corresponding revisions in the manuscript.

  1. Instrument validation – The tasks used remain unvalidated, with no pilot testing or reliability evidence provided.
    Response: We agree with the reviewer that the absence of validated tools is a major methodological limitation of our study. We have now explicitly acknowledged this in the Introduction (Section 1.5, “Rationale and Aim”) and in the Limitations and Future Directions section of the Discussion. In the Introduction, we added that no standardized battery exists for assessing categorical liaison, and we explicitly describe our tasks as ad hoc and exploratory rather than confirmatory.
  2. Task variability – The inclusion of multiple task formats introduces uncontrolled variability that is not sufficiently addressed.
    Response: We have now expanded the Materials and Methods section (2.2) to justify our decision to include diverse task formats. We explain that this was a deliberate design choice to maximize ecological validity and maintain child engagement, while clearly acknowledging that such diversity introduces methodological heterogeneity and that no formal reliability coefficients could be computed. We also emphasize that the results should be interpreted as preliminary and hypothesis-generating.
  3. Cognitive measures – The Flanker task lacks calibration for this age group and clinical population.
    Response: This point has been explicitly incorporated into both the Methods (Section 2.2) and Limitations section of the Discussion, where we now state that the Flanker task is not specifically normed for children with expressive DLD, and that this limits the strength and generalizability of our conclusions regarding inhibitory control.
  4. Design – There is still no randomization or counterbalancing of task administration, raising concerns about fatigue and order effects.
    Response: We have now acknowledged this limitation explicitly in the Limitations and Future Directions section of the Discussion and noted that future studies should employ randomization or counterbalancing to better control for order and fatigue effects.
  5. Statistical analysis – The reliance on Mann–Whitney U tests and correlations remains a limitation, particularly given the ambitious theoretical claims.
    Response: We have clarified in the Data Analysis section (2.4) why non-parametric tests were used (small and heterogeneous sample), and we explicitly state in the Discussion that the results are descriptive and hypothesis-generating, not confirmatory, and should be interpreted with caution. We also note that future work should employ more robust analyses (e.g., linear mixed-effects models, MANOVA) with larger cohorts.
  6. Overall framing – The study should be positioned more clearly as exploratory/pilot.
    Response: We have systematically reframed the manuscript as exploratory throughout:
  • The title now specifies that this is an “Exploratory Study.”
  • The abstract has been revised to indicate that the findings are preliminary and hypothesis-generating.
  • The Introduction (end of Section 1.5) explicitly frames the study as exploratory and describes its goal as providing descriptive groundwork for future research.
  • The Discussion and Conclusion have been revised to moderate theoretical claims and emphasize the hypothesis-generating nature of the findings.

Reviewer 2 Report (New Reviewer)

Comments and Suggestions for Authors

This is an interesting study on categorical liaison of French children with DLD. The study is well-designed and the findings are presented clearly. There are some minor issues for the authors to consider:

  • Abstract: The definition of categorical liaison should be provided before the introduction of the current project as it lays the foundation of the paper.
  • Line 17: Mastery of categorical liaison is a marker of DLD? Please revise.
  • Lines 32-33: If the references are listed according to their order of occurrence, why are you using [1, 83, 84]?
  • Line 69: “In addition to the lexicon, syntax and morphology…” is a bit confusing regarding. It’s better to separate morphology and syntax in two paragraphs and revise the wording.
  • Line 84: A brief introduction to categorical liaison is expected for readers unfamiliar with this feature. Or a sentence directing readers to the next section is preferred.
  • Lines 96-107: Please state if these are exclusive cases.
  • Lines 388-389: It can be omitted, as it is standard convention in statistical reporting.
  • Lines 584-592: The authors may elaborate more on the limitations and future directions.
Comments on the Quality of English Language

The authors may have the manuscript proofread by a native professional.

Author Response

Response to Reviewer 2

We sincerely thank Reviewer 2 for their careful reading of our manuscript and for their constructive suggestions. Their comments have helped us improve the clarity and precision of the text and ensure that the scope of the study is appropriately framed. Below, we provide a point-by-point response indicating how each comment was addressed.

  1. Abstract: “The definition of categorical liaison should be provided before the introduction of the current project as it lays the foundation of the paper.”
    Response: We fully agree. The abstract has been revised to provide a brief and precise definition of categorical liaison at the very beginning, before introducing the aims and findings of the study.
  2. “Line 17: Mastery of categorical liaison is a marker of DLD? Please revise.”
    Response: This sentence was rewritten in the abstract and Introduction to clarify that persistent difficulty with categorical liaison production (beyond age 5) may serve as a clinical marker of expressive DLD, rather than mastery being a marker.
  3. “Lines 32–33: If the references are listed according to their order of occurrence, why are you using [1, 83, 84]?”
    Response: We have revised the reference numbering throughout the manuscript to ensure strict adherence to order of appearance. The first three references now appear as [1–3] (national prevalence data), followed by [4–5] (international estimates).
  4. “Line 69: ‘In addition to the lexicon, syntax and morphology…’ is a bit confusing.”
    Response: We revised this passage in Section 1.1 (Clinical Approach) to separate morphology and syntax into distinct paragraphs and to clarify the explanation of morphosyntactic impairments.
  5. “Line 84: A brief introduction to categorical liaison is expected for readers unfamiliar with this feature.”
    Response: We have significantly expanded Section 1.2 (Linguistic Perspective) to provide a detailed introduction to categorical liaison, including a list of cases where liaison is obligatory and an explanation of why these contexts are important for psycholinguistic research.
  6. “Lines 96–107: Please state if these are exclusive cases.”
    Response: We have clarified that our study focused primarily on determiner–noun contexts and subject–verb–determiner–noun frames in perception tasks, excluding optional and illegal liaisons, and we explain why these choices were made (to ensure experimental control and avoid ambiguity).
  7. “Lines 388–389: It can be omitted, as it is standard convention in statistical reporting.”
    Response: The sentence in question was removed from the Results section.
  8. “Lines 584–592: The authors may elaborate more on the limitations and future directions.”
    Response: We have substantially expanded the Limitations and Future Directions section of the Discussion to include (a) sample size constraints, (b) absence of randomization/counterbalancing, (c) lack of validated tools and task reliability metrics, (d) limitations of the Flanker task for this age group, and (e) the need for more advanced statistical models in future work.
  9. “The authors may have the manuscript proofread by a native professional.”
    Response: The entire manuscript was carefully revised for grammar, clarity, and fluency.

Reviewer 3 Report (New Reviewer)

Comments and Suggestions for Authors

Title: Production and Processing of the French Categorical Liaison in
Children with Expressive Developmental Language Disorder: Behavioral Data
from a Cross-Sectional Study

Many thanks for the invitation and this topic is very important for the scientific community. I observed that revisions were made already with origin in previous rounds, but some limitations can be completed. I wrote my recommendations below.

- regarding the sample and representativeness, the study included only 24 children with expressive DLD and 22 controls, limiting statistical power and generalizability. Please be sure that you develop argument to acknowledge this in limitations section;

- the heterogeneity in age is rarely observed along the literature review (6–10 years) as well educational level (not accurately assessed for the DLD group) what suggests some bias regarding uncontrolled variance that may obscure age-related trends. But authors can explain this as a limitation / or develop a new paragraph, in Introduction, to compare with % in other age groups and educational levels in France and also considering a broader cross-cultural perspective (this is an article, not a data report);

  • explain more about the properties (validity) of the tests used in this sample;
  • pay attention: the absence of standardized norms for the Flanker task in this specific age group with expressive DLD undermines the robustness of conclusions regarding inhibition;
  • about different competencies: reading ability was used to classify participants as “readers” and “pre-readers,” but literacy was neither quantified nor controlled beyond this dichotomy. Just address some explanation in this case;
  • also as limitations to be identified by authors: the study did not account for socioeconomic status, bilingual exposure, or comorbid conditions that could influence language development and executive functioning; in fact I would suggest more literature that is relevant in this topic;
  • given the small sample, non-parametric Mann–Whitney tests were used, but more nuanced analyses (e.g., linear mixed-effects models) could account for individual variability and interactions between factors such as age, reading level, and memory measures. Authors can explain, in data analysis, why their decision regarding the tests used for statistics;
  • no correction for multiple comparisons was reported, increasing the risk of Type I errors – this can be ‘saved’ if acknowledged in limitations’ section;
  • in results: significant correlations between working memory/inhibition and liaison production do not establish causality; and the interpretation that procedural memory deficits underlie categorical liaison difficulties remains speculative without direct neurocognitive measures. Please assure robustness for explanation here;

Many thanks. Please use track changes during your revision in the manuscript and a letter confirming the recommendations if they attended.

Author Response

Response to Reviewer 3

We are very grateful to Reviewer 3 for the thoughtful and detailed comments, which encouraged us to clarify sample characteristics, acknowledge potential sources of variability, and better justify our methodological decisions. Below, we outline the revisions made in response to each of these points.

  1. “The study included only 24 children with expressive DLD and 22 controls, limiting statistical power and generalizability.”
    Response: We agree with this important observation. The Limitations and Future Directions section was expanded to explicitly acknowledge the small sample size as a limitation that reduces statistical power and generalizability. We have also indicated that replication with larger and more diverse samples is needed to confirm these findings.
  2. “The heterogeneity in age (6–10 years) and educational level may obscure age-related trends.”
    Response: We have added a paragraph in the Introduction (Section 1.1) describing the heterogeneity of age and educational delay in children with DLD, clarifying that such heterogeneity is frequently observed in inclusive schooling contexts in France. We explain that, although this increases within-group variance, it reflects the real-world educational profiles of children with expressive DLD and provides ecologically valid data. We also emphasize this as a limitation in the Discussion.
  3. “Explain more about the properties (validity) of the tests used in this sample.”
    Response: We have expanded the Materials and Methods (Section 2.2) to specify that our tasks were partly adapted from previously published studies (e.g., Chevrot et al., Wauquier, Dugua) and partly newly designed to suit our clinical population. We clearly state that these tasks have not been formally standardized or validated, and that our results should be considered exploratory and hypothesis-generating.
  4. “Absence of standardized norms for the Flanker task in this population undermines conclusions about inhibition.”
    Response: This concern is now explicitly mentioned as a limitation in the Discussion, and we call for the development and validation of age-appropriate inhibitory control tasks for children with DLD in future research.
  5. “Reading ability was used to classify participants as ‘readers’ and ‘pre-readers,’ but literacy was not quantified or controlled.”
    Response: We have added a clarifying note in Methods (Participants section) to explain that reading classification was based on school records and teacher reports, but that no standardized literacy measure was administered. This limitation is also acknowledged in the Discussion.
  6. “Socioeconomic status, bilingual exposure, and comorbidities were not accounted for.”
    Response: We have explicitly stated in the Limitations section that these factors were not systematically controlled and that they may have influenced language and executive function outcomes. We recommend that future studies collect this information systematically.
  7. “More nuanced analyses (e.g., linear mixed-effects models) could account for variability and factor interactions.”
    Response: In Data Analysis, we now justify our reliance on nonparametric statistics, explaining that the small sample size, non-normal distributions, and heterogeneous variance made Mann–Whitney U tests the most robust choice. We acknowledge in the Limitations section that more advanced modeling (e.g., mixed-effects regression) would be valuable in future studies with larger samples.
  8. “No correction for multiple comparisons was reported.”
    Response: This point is acknowledged in the Limitations section, and we note that our exploratory approach prioritizes hypothesis generation, which further supports the need for replication in larger confirmatory studies.
  9. “Correlations between working memory/inhibition and liaison production do not establish causality.”
    Response: We now explicitly state in the Discussion that these relationships are correlational and should be interpreted with caution. We also recommend that future longitudinal studies and training interventions explore causal links more directly.

Round 2

Reviewer 1 Report (Previous Reviewer 2)

Comments and Suggestions for Authors

All comments have been well addressed. I have no more comments.

This manuscript is a resubmission of an earlier submission. The following is a list of the peer review reports and author responses from that submission.

Round 1

Reviewer 1 Report

Comments and Suggestions for Authors

Abstract:

This study addresses a relevant and timely topic: the production of liaison in French. However, the authors should explicitly state that the study focuses on oral production, as there is also a body of research examining written liaison in French (e.g., Christiane Soum's work), which could otherwise lead to confusion. Additionally, the abstract lacks clarity regarding the linguistic processes assessed. The authors mention comprehension, production, and identification, yet these processes are not interchangeable. It is important to clearly distinguish them so that the scope and nature of the assessment are fully understood by the reader. Finally, the abstract would benefit from explicitly stating the main research objective at the outset, in order to clearly frame the study and its contribution.

Introduction :

While the authors state that their study focuses on expressive dysphasia, they do not provide a clear justification for choosing this specific subtype over others. Additionally, the prevalence of expressive dysphasia is not explicitly reported, which weakens the rationale for focusing solely on this form. Given that the introduction mentions three distinct types of dysphasia (receptive, mixed, and expressive), it would be important for the authors to clarify why expressive dysphasia was selected for investigation, particularly whether it is due to its frequency, specific linguistic features, or methodological feasibility. Including prevalence data and a brief comparative rationale would strengthen the argument for the study's scope and relevance.

The authors might consider providing a more detailed explanation as to why the production of liaison in spoken French proves particularly challenging for individuals with dysphasia. Specifically, it would be beneficial to determine whether this difficulty arises solely from phonological impairments or if syntactic and morphological deficits also play a significant role. Moreover, clarifying whether this issue uniformly affects all subtypes of dysphasia (expressive, receptive, mixed) or manifests differently across them would enhance the depth of the analysis. Additionally, it would be valuable to investigate whether liaison difficulties extend to written language production or if this aspect remains relatively intact. Such a comparison between oral and written modalities could yield important insights into the underlying mechanisms of the linguistic impairments associated with dysphasia.

The paragraphes (« The present study aimed to fill this gap » l.106-128 ; « Consequently, the categorial liaison constitutes a powerful » l.154-161) are inappropriately placed within the middle of the theoretical introduction. It would be more effective to position it at the conclusion of the introduction section, thereby providing a clear articulation of the research problem and rationale for the study. Reorganizing this content to serve as a problem statement would enhance the logical coherence and overall clarity of the manuscript.

The entire section on memory appears overly lengthy and could be streamlined. It would be more effective to retain only the essential points, particularly those directly relevant to discussing the study’s results, namely, the key studies linking memory systems with dysphasia. This focused approach would enhance clarity and strengthen the discussion without diluting the core arguments.

The section 1.5. outlining the targeted developmental period (ages 6 to 10) for the study lacks a clear scientific justification for the selection of this specific age range. It would be beneficial for the authors to explicitly state the theoretical framework or developmental model underpinning this choice. For instance, do they base it on previous findings related to key stages in the acquisition of categorical liaison in French, critical periods in cognitive or language development, or specific milestones concerning the maturation of working memory and executive control? Without such contextualization, the choice of this age range appears somewhat arbitrary, which limits the interpretability and relevance of the study’s findings.

Based on the hypothesized effect sizes and the factorial design involving group (dysphasia vs neurotypical), age subgroups, and task conditions, a power analysis suggests that a minimum of 30-40 participants per group would be necessary to reliably detect medium-sized effects and interactions with a power of 0.80 at an alpha level of 0.05. Given the complexity of the design and the inclusion of cognitive covariates, the current sample size of 24 dysphasic and 22 neurotypical children may be underpowered for detecting subtle interaction effects, especially those involving developmental factors. Increasing the sample size would strengthen the statistical validity and generalizability of the findings.

The matching between the dysphasic and neurotypical children groups relies solely on chronological age, which represents a significant limitation. Indeed, to ensure the validity of comparisons, it would have been necessary to assess at least the abbreviated IQ of the neurotypical children to achieve a rigorous match on cognitive abilities. Furthermore, in studies on dysphasia, common matching criteria include not only age but also standardized language measures and cognitive indicators, which help isolate the effect of the language disorder more precisely. Additionally, the gender composition of samples must be taken into account, as dysphasia shows a markedly higher prevalence in boys, with an estimated male-to-female ratio between 3:1 and 4:1. Statistical control of gender is therefore essential to avoid bias arising from this prevalence difference. Incorporating these recommendations would strengthen the reliability of conclusions drawn regarding differences between dysphasic and neurotypical children.

Reviewer 2 Report

Comments and Suggestions for Authors
  1. The manuscript investigates a linguistically significant issue—categorical liaison in children with expressive dysphasia, and includes a range of verbal and non-verbal tasks that are clearly described in Section 2.2. However, the study lacks formal validation of these instruments. While the structure and content of each task are outlined, the absence of reliability data, construct validation, or piloting, especially for a clinical child population, remains a serious limitation.
  2. The diversity of production task types (naming, enumeration, riddles, storytelling, guided conversation) introduces uncontrolled variability in cognitive and linguistic demands. Without clear standardization or explanation of how performance was equated across task types, it becomes difficult to attribute observed effects to dysphasia rather than to task format or complexity.
  3. The cognitive measures (digit span and Flanker test) are well-established, but the manuscript does not provide evidence of their calibration for this age range or diagnostic group. This is particularly important for the Flanker task, which requires sustained attention and executive functioning that may be unevenly developed in children with language impairments.
  4. There is no indication that test administration was randomized or counterbalanced across participants. Given the vulnerability of the population to fatigue and task-order effects, this is a significant design flaw that could bias the results.
  5. The statistical analysis is limited to Mann-Whitney U tests and correlations. This may be appropriate for a small sample, but the theoretical claims, particularly those related to declarative and procedural memory, inhibitory control, and schema acquisition, require more robust analytic support. Multivariate models or even hierarchical regressions would be more appropriate given the layered nature of the data (group × age × task type).
